# Supplementation with Magnesium Salts—A Strategy to Increase Nutraceutical Value of *Pleurotus djamor* Fruiting Bodies

**DOI:** 10.3390/molecules26113273

**Published:** 2021-05-28

**Authors:** Piotr Zięba, Agnieszka Sękara, Emilia Bernaś, Agata Krakowska, Katarzyna Sułkowska-Ziaja, Edward Kunicki, Małgorzata Suchanek, Bożena Muszyńska

**Affiliations:** 1Department of Horticulture, Faculty of Biotechnology and Horticulture, University of Agriculture in Kraków, 29 Listopada 54, 31-425 Kraków, Poland; p.zieba90@gmail.com (P.Z.); edward.kunicki@urk.edu.pl (E.K.); 2Department of Plant Product Technology and Nutrition Hygiene, Faculty of Food Technology, University of Agriculture in Krakow, 122 Balicka Street, 30-149 Kraków, Poland; emilia.bernas@urk.edu.pl; 3Department of Inorganic and Analytical Chemistry, Faculty of Pharmacy, Jagiellonian University Medical College, 9 Medyczna Street, 30-688 Kraków, Poland; agata.krakowska@uj.edu.pl; 4Department of Pharmaceutical Botany, Faculty of Pharmacy, Jagiellonian University Medical College, 9 Medyczna Street, 30-688 Kraków, Poland; katarzyna.sulkowska-ziaja@uj.edu.pl (K.S.-Z.); muchon@poczta.fm (B.M.); 5Department of Analytical Chemistry and Biochemistry, Faculty of Materials Science and Ceramics, AGH University of Science and Technology, Al. Mickiewicza 30, 30-059 Kraków, Poland; msuchanek@agh.edu.pl

**Keywords:** pink oyster, *Pleurotus djamor*, functional food, biofortification, magnesium

## Abstract

The use of substrates supplemented with minerals is a promising strategy for increasing the nutraceutical value of *Pleurotus* spp. The current research was performed to analyze the effect of substrate supplementation with magnesium (Mg) salts on the Mg content, biomass, and chemical composition of pink oyster mushroom (*Pleurotus djamor*) fruiting bodies. Before inoculation, substrate was supplemented with MgCl_2_ × 6 H_2_O and MgSO_4_, both salts were applied at three concentrations: 210, 420, and 4200 mg of Mg per 2 kg of substrate. The harvest period included three flushes. Substrate supplementation with 4200 mg of Mg caused the most significant decrease in mushroom productivity, of about 28% for both Mg salts. The dry matter content in fruiting bodies was significantly lower in the treatment in which 210 mg of Mg was applied as MgSO_4_ in comparison to the control. Supplementation effectively increased the Mg content in fruiting bodies of *P. djamor* by 19–85% depending on the treatment, and significantly affected the level of remaining bioelements and anions. One hundred grams of pink oyster fruiting bodies, supplemented with Mg salts, provides more than 20% of the Mg dietary value recommended by the Food and Drug Administration (FDA); thus, supplementation can be an effective technique for producing mushrooms that are rich in dietary Mg. Although *P. djamor* grown in supplemented substrate showed lower productivity, this was evident only in the fresh weight because the differences in dry weight were negligible. Mg supplementation increased the antioxidant activity of the fruiting bodies, phenolic compounds, and some amino acids, including L-tryptophan, and vitamins (thiamine and l-ascorbic acid).

## 1. Introduction

The species of the genus *Pleurotus*, commonly known as oyster mushrooms, are distributed in tropical and temperate regions of the world. Some of these, including *P. citrinopileatus, P. djamor*, *P. eryngii*, *P. flabellatus*, *P. florida*, and *P. ostreatus* have been used as both food and medicine since the time of ancient civilizations. They have now gained wide popularity as nutraceuticals due to their high nutritional and medicinal value and their ability to grow on commonly available agricultural wastes [1,2]. Nutritionally, oyster mushrooms are valued as sources of protein, fiber, carbohydrates, vitamins, and minerals. Nagy et al. [3] reported 17.9–30.3% dry weight (DW) of crude proteins, 1.10–2.2% DW of lipids and 57.6–62.4% DW carbohydrates in *Pleurotus ostreatus* fruiting bodies collected from natural habitats. The major amino acids were lysine and valine, and fatty acids determined at the highest amounts were linoleic and oleic acids. Oyster mushrooms contain biologically active substances—immuno-modulating polysaccharides, indole compounds, and numerous antioxidants, such as phenolic acids or ergothioneine [4,5]. The amino acid score of *P. ostreatus* meets the nutritional requirements of all essential amino acids for adults, with the one limiting amino acid, leucine, which can be easily acquired by supplementing with cereals [6,7]. Thus, oyster mushrooms are considered to be a functional food, which should be a standard component of the human diet [8]. Many medicinal effects have been confirmed for *Pleurotus* spp.; as a result, so mycelia and fruiting bodies are used as raw materials by the pharmaceutical and cosmetics industries [5,9]. Researches has also confirmed positive health and potential medicinal properties of basidiocarps and mycelium obtained from in vitro cultures of *P. djamor*. β-glucans isolated from pink oyster mushroom have a cytotoxic effect against ovarian carcinoma cells in in vitro models [10]. The β-glucans complex with zinc obtained from *P. djamor* mycelium has an antioxidative impact on liver and kidney damage [11,12,13]. Acetylated polysaccharides obtained from in vitro mycelium have anti-aging effects in D-galactose-induced aging mice, suggesting potential application to age-related diseases in human [14]. 

*Pleurotus djamor* (Rumph. ex Fr.) Boedijn is a tropical species of the oyster mushrooms group native to South East Asia and Central America [15]. In tropical regions, pink oyster mushrooms are commonly collected in the wild and are readily available in rural markets. These mushrooms are appreciated and eagerly bought by customers as a substitute for soybean or egg [16]. Although this species is less important economically than the grey oyster mushroom (*P. ostreatus*), it is becoming more popular due to its unusual pink color and specific texture that resembles fried meat. *P. djamor* can be easily grown on various agriculture and food residues [17,18]. Furthermore, it is more disease-resistant than the grey oyster mushroom and can be cultivated at higher temperatures, even exceeding 35 °C [19]. The main challenge of pink oyster mushroom cultivation and trade is the mushroom’s short shelf life. Basidiocarps have to be picked when they are young; before spores’ release, they cannot be stored for more than a few days [15]. 

Production of the mushrooms is based on agricultural and agro-industrial wastes. The residual substrate after mushroom harvesting can be used as a valuable organic fertilizer or a complement for animal feed. Therefore, mushroom production meets the requirements of modern, sustainable agriculture, while simultaneously providing a functional food [20,21,22]. Mushrooms produce several enzymes that degrade lignin, hemicellulose, and cellulose. Lignin–cellulose substrates are widely used in mushroom cultivation [23]. Mushroom mycelium converts lignocellulosic residues into a food product in oxygen conditions and at a specific pH, gaining all macroelements (C, N, P, K, and Mg), and microelements (Fe, Se, Zn, Mn, Cu, and Mo) from a substrate, which is sometimes supplemented with an additional source of N [24,25]. In addition to supplementing the growing substrate using protein, several authors have underlined the role of supplements containing fats, carbohydrates, fibers, and ashes to increase mushroom yields [26]. Alternatively, substrate supplementation in mineral salts was recently recognized as an effective method of enriching the bioelements of mycelia and fruiting bodies. Oliveira and Naozuka [27] demonstrated that *P. djamor* accumulates selenium and Se-proteins, which were reported to be highly bio-accessible. Włodarczyk et al. [28] stated that the addition of Zn and Mg salts’ to the media increased the biomass production of *Pleurotus* spp. mycelium and the accumulation of inorganic salts. Mg is an essential macroelement in the human diet, with a vital role in numerous metabolic processes. Its deficit is often encountered in many countries, particularly among older people [29,30,31]. Considering the above, the biofortification of oyster mushrooms with Mg could be an alternative means of producing a functional food.

This research aimed to determine the impact of fortification of the substrate with magnesium salts the productivity and chemical composition of *P. djamor* fruiting bodies. To the best of our knowledge, this issue has not been sufficiently described in the scientific literature, despite its potential scientific importance and applicability to the horticulture and food production sectors. From a practical perspective, this research contributes to developing sustainable horticulture, in addition to improving competitiveness and increasing the incomes of the mushroom production sector. The hypotheses verified within this research include: (i) the dose-dependent increase in Mg content in fruiting bodies of *P. djamor* as a result of the addition of Mg salts to the substrate; (ii) the differentiated effect of supplementation with Mg chloride and sulfide on the Mg level in mushroom fruiting bodies; (iii) dose- and salt-dependent effects of Mg supplementation on the biomass of fruiting bodies; and (iv) salt-dependent effects of Mg supplementation on the chemical composition of fruiting bodies.

## 2. Materials and Methods

### 2.1. Mushroom Materials and Experiments Design

The material used in this study was *Pleurotus djamor* (Rumph. ex Fr.) Boedijn maintained in agar culture, from the deposit of the Department of Horticulture, University of Agriculture in Krakow, Poland. 

The substrate prepared for *P. djamor* cultivation was supplemented with magnesium salts: magnesium chloride hexahydrate (MgCl_2_·6H_2_O) and magnesium sulfate (MgSO_4_) both from Warchem (Warszawa, Poland). These magnesium salts (MgCl_2_·6H_2_O and MgSO_4_) we applied to the substrate at three concentrations (i) 50% of dietary value (DV) for Mg, namely 210 mg of Mg in a form of chloride (MgCl_2_ × 0.5) and sulfate (MgSO_4_ × 0.5) per 2 kg of substrate, (ii) 100% of DV for Mg, namely 420 mg of Mg (MgCl_2_ × 1 and MgSO_4_ × 1); and (iii) 1000% of DV for Mg, namely 4200 mg of Mg (MgCl_2_ × 10 and MgSO_4_ × 10). The DV used was based on the 2015–2020 Dietary Guidelines for Americans for Mg, equal to 420 mg per day for adult men [32]. Thus, the experimental layout was as follows: two Mg salts (first source of variation) and three concentrations of both Mg salts (additional source of variation).

### 2.2. Substrate Preparation

In the first stage of the experiment, rye grain (from local farmer) mycelium was prepared. Hydrated rye was placed in polypropylene bags with a microfilter and sterilized at 121 °C for 1.5 h at 1 atm pressure in an ASVE steam sterilizer (SMS, Poland). After cooling, grain was inoculated with fresh *P. djamor* mycelium, and incubated at 23 °C, in darkness until the rye grain was fully colonized with mycelium. The cultivation substrate was prepared from hardwood beech pellets (Biomasa Magdalena Małaczyńska, Trześń, Poland) and wheat bran (from local farmer) in a ratio of 5:1 (*v*:*v*) with the addition of 1% dry weight (DW) of gypsum, which is the standard cultivation medium used in our laboratory. Substrate was hydrated to obtain a moisture level of 65%, mixed, and placed in polypropylene bags with a microfilter in an amount of 2 kg of substrate per bag. The magnesium salts were added to each bag in the concentrations described above and mixed with substrate (three bags per combination of salts). Substrate in bags was sterilized for 1.5 h at 121 °C and pressure of 1 atm in an ASVE steam sterilizer (SMS, Poland). The cooled bags were inoculated with 3% rye grain mycelium obtained in the first stage, and the entire mix was molded into cubes, which were incubated at 23 ± 1 °C. After being fully colonized by mycelium, the cultivation cubes were placed in a cultivation chamber, in which constant growing conditions were maintained: 90 ± 3% humidity, 18 ± 2 °C, and a photoperiod of 12 h of light intensity of 11 μmol s^−1^ m^−2^. Fruiting bodies were collected when they reached the harvesting maturity, which meant before the spores’ release. Mushrooms were harvested in 3 flushes—the first flush was used for fresh material analyzed; all homogenized flushes were used for dry weight analysis. 

### 2.3. Dry Weight and Biological Efficiency

Mushrooms were weighed with a Sartorius A120S balance (Sartorius AG, Göttingen, Germany) to determine the fresh weight (FW) and oven dried at 65 °C until a constant weight was obtained to determine the dry weight. Then, the difference in weight was calculated and expressed as the FW percentage. Biological efficiency was calculated by relating the total harvest from 3 flushes to 1000 g of dry substrate (weight of harvest/weight of dry substrate) × 100%.

### 2.4. Determination of Total Phenols Content

The concentration of total phenols in mushroom extracts was estimated by the Folin–Ciocalteu colorimetric procedure described by Djeridane et al. [33] with modifications. Two grams of fresh mushroom material was mixed with 10 mL of 80% methanol (Chempur, Gliwice, Poland) and centrifuged for 10 min at 3492 g. Next, mushroom extracts (0.1 mL) were mixed with 2 mL of 2% sodium carbonate (Na_2_CO_3_) (Warchem, Warszawa, Poland); after 2 min, 0.1 mL of Folin–Ciocalteu reagent (Sigma-Aldrich, Darmstadt, Germany), mixed with deionized water (1:1 *v*/*v*), was added to the test tubes. The final mixture was shaken and then incubated for 45 min in the dark at room temperature before the absorbance was measured at 750 nm using an ultraviolet–visible (UV-VIS) Helios Beta spectrophotometer (Thermo Fisher Scientific Inc., Waltham, MA USA) against a reference solution containing 0.1 mL of methanol instead of 0.1 mL of mushroom extract. The results were determined from a standard gallic acid curve and are expressed as milligrams of gallic acid equivalents (GAE) per gram FW (mg GAE g^−1^ FW).

### 2.5. Determination of Antioxidant Activity Using DPPH^•^ (2,2-Diphenyl-1-picrylhydrazyl) Method

The antioxidant activity (AA) was determined using 2,2-diphenyl-1-picrylhydrazyl (DPPH^•^) (Sigma-Aldrich, Darmstadt, Germany) as a free radical [34]. The decrease in absorbance was measured at 517 nm with an UV-VIS Helios Beta spectrophotometer. Mushroom extracts used in total phenols content were used. The test tubes contained 0.1 mL of supernatant and 4.9 mL of 0.1 mM DPPH^•^ dissolved with 80% methanol. The reaction mixture was shaken and incubated in the dark at 20 °C for 15 min. The following formula was used to calculate DPPH^•^ radical scavenging activity: AA (%) = ((A0 − A1)/A0) × 100; AA—antioxidant activity, A0—absorbance of the reference solution, and A1—absorbance of the test solution. AA was expressed as the percentage of DPPH^•^ free radical scavenging

### 2.6. Total Soluble Sugars

Total soluble sugars were determined using the anthrone method described by Yemm and Willis [35]. Fresh mushroom material (10 g) was mixed with 80% ethanol (Warchem, Warszawa, Poland) and anthrone reagent (Sigma-Aldrich, Darmstadt, Germany), then the absorbance was measured at 625 nm with a Helios Beta spectrophotometer.

### 2.7. L-Ascorbic Acid Content

The content of L-ascorbic acid was determined by Tillman’s titration method as described by Krełowska-Kułas [36]. Fresh mushroom material (12.5 g) was homogenized with 50 mL acetic acid (Warchem, Warszawa, Poland) applied as an acidity regulator. After 30 min, the mixture was titrated with Tillman’s reagent (2,6-dichlorophenol-indophenol) (Sigma-Aldrich, Darmstadt, Germany). Excessive dye in an acidic environment gives a pink color and marks the end point of the titration. Then, the content of L-ascorbic acid in the sample was calculated based on the amount of the changed solution of 2,6-dichlorophenol-indophenol used for titration.

### 2.8. Glutathione Content

The reduced form of glutathione (GSH) was determined according to the method described by Guri [37] with some modifications. Fresh mushroom material (2.5 g) was homogenized in an ice bath (4 °C) with the addition of 6 mL of 0.5 mM ethylenedinitrilotetraacetic acid (EDTA) and 3% trichloroacetic acid (TCA) both from Sigma-Aldrich (Darmstadt, Germany), and the homogenate was centrifuged at 4 °C for 10 min at 6208 g. To bring the pH of the solution to ca. 7.0, K-phosphate buffer (Pol-Aura, Dywity, Poland) was used. The content of reduced GSH was assessed using Ellman’s reagent (5,5-dithiobis-2-nitrobenzoic acid, DTNB) (Sigma-Aldrich, Darmstadt, Germany) on a Helios Beta spectrophotometer. The solution extinction was measured at the wavelength of *λ* = 412 nm. The absorbance of a mixture of 2.0 mL of mushroom homogenate and 1 mL of 0.2 M K-phosphate buffer, which absorbed part of the radiation was measured as a blind sample. The concentration of GSH was calculated from the standard curve and expressed as μg g^−1^ FW.

### 2.9. Analysis of Bioelements

Dried fruiting bodies were powdered in an Pulverisette 14 ball mill (Fritsch GmbH, Idar-Oberstein, Germany; 0.5 mm sieve). Samples were analyzed for the content of Mg, K, Ca, Fe, Zn, and Cu. Three independent samples (0.2 g) were weighed from each of the dried mushroom materials and were transferred into Teflon vessels containing 2 mL of 30% H_2_O_2_ and 6 mL 65% HNO_3_ both from Merck (Darmstad, Germany). Then, the samples were subjected to wet mineralization in a closed system in a Magnum II mineralizer (ERTEC, Wrocław, Poland). The mineralized solution obtained was heated on a hot-plate for 60 min at 120 °C to remove excess reagents. Subsequently, all the samples were quantitatively transferred to 10 mL flasks and topped with quadruple distilled water. To determine elements, flame atomic absorption spectrometry (FAAS) was used. For all the measurements, an atomic absorption spectrometer by Thermo Scientific (Model iCE 3500, Cambridge, UK) was used.

### 2.10. Determination of Chloride and Sulfate Ions

Cl^−^ and SO_4_^2−^ were determined using spectrophotometry with a Spectroquant Nova 60 spectrophotometer (Merck KGaA, Darmstadt, Germany). We used validated tests to determine SO_4_^2−^ (Cat. No. 101812, Merck KGaA, Darmstadt, Germany) and Cl^−^ (Cat. No. 114897, Merck KGaA, Darmstadt, Germany). Spectrophotometric determinations were performed in quartz cuvettes. The results of the determination of the content of Cl^−^ and SO_4_^2−^ anions are presented as mean values from three independent measurements.

### 2.11. Analysis of Organic Compounds—Preparation of the Extract

Dried fruiting bodies were powdered in an Pulverisette 14 ball mill. Three grams of mushroom material was extracted with methanol by ultrasound (49 kHz for 30 min; Sonic-2, Polsonic, Warszawa, Poland). The extraction was repeated thrice for each sample. The obtained extracts were combined (300 mL) and evaporated to dryness. Subsequently, the extracts were quantitatively dissolved in high-performance liquid chromatography (HPLC)-grade methanol and filtered using membrane filters.

#### 2.11.1. Determination of Phenylalanine and Phenolic Acids

The analysis of the content of phenylalanine and phenolic acids in the tested samples was carried out according to the procedure proposed by Sułkowska-Ziaja et al. [38]. The analysis was performed using reversed-phase high-performance liquid chromatography (RP-HPLC VWR, Hitachi-Merck, Tokyo, Japan) equipped with a DAD (diode array detector) (L-2455) (*λ* = 254 nm) and autosampler (L-2200), pump (L-2130), and RP-18e LiChrospher column (4 × 250 mm, 5 µm) kept at 25 °C. The mobile phase was prepared as follows: solvent A: methanol/0.5% acetic acid 1:4 (*v*/*v*) and solvent B: methanol. The gradient was set as follows: 100:0 time 0–25 min; 70:30 time 35 min; 50:50 time 45 min; 0:100 time 50–55 min; 100:0 time 57–67 min. The identification of the compounds was undertaken by comparing the obtained spectra with the spectra of the standard compounds (purity ≥99.0% from Sigma-Aldrich) and the compounds of the spectrum were compared. Quantitative analysis was performed using a calibration curve. The content of compounds is expressed in mg/100 g dry weight.

#### 2.11.2. Analysis of Indole Compounds

The extracts were analyzed for the content of indole compounds using the RP-HPLC method with UV detection [39]. The prepared extracts were quantitatively dissolved in 1.5 mL of the solvent mixture and the components were separated via RP-HPLC method (Hitachi RP-HPLC with UV detection, Merck, Tokyo, Japan) equipped with an L-7100 type pump. The Purospher^®^ RP–18 column (4 × 200 mm, 5 µm) was maintained at 25 °C and equipped with a UV detector (*λ* = 280 nm). The applied liquid phase consisted of a mixture of methanol/water/ammonium acetate (Chempur, Gliwice, Poland) (15:14:1 *v*/*v*). The flow rate was established at 1 mL min^−1^. Indole compounds were quantitatively analyzed with the help of a calibration curve and with the assumption of linearity of the size of the area tested under the peak relative to the concentration of the standard used (purity ≥99.0% from Sigma-Aldrich, St. Louis, MO, USA).

#### 2.11.3. Analysis of Lovastatin

To determine the content of lovastatin the RP-HPLC method was used according to the method described by Pansuriya and Singhal [40]. The process was carried out in an isocratic system with a mobile phase of constant composition. The apparatus was equipped with a UV detector (*λ* = 238 nm), a column (Purospher^®^ RP-18, 14 × 200 mm, 5 µm), and a lamp (L-7100). All the measurements were carried out using a previously prepared developing system (acetonitrile and 0.1% phosphoric acid at a ratio of 60:40 *v*:*v*) (Chempur, Gliwice, Poland). Lovastatin was quantitatively analyzed with the help of a calibration curve and with the assumption of linearity of the size of the area tested under the peak relative to the concentration of the compound standard (purity ≥99.0% from Sigma-Aldrich, St. Louis, MO, USA) used.

### 2.12. Determination of Vitamin B_1_ (Thiamine) and B_2_ (Riboflavin) Content

The content of vitamins B_1_ and B_2_ in freeze-dried *P. djamor* was determined by the HPLC method [41,42]. The aforementioned vitamins were determined after the oxidation reaction before the column using potassium hexacyanoferrate(III) solution, purity ≥99.0% from Sigma-Aldrich, Merck KGaA, Darmstadt, Germany. Later, reaction and centrifugation extracts were cleaned on solid phase extraction (SPE) using Chromabond C18 columns (3 mL/200 mg). Finally, 20 µL of sample was injected into the HPLC system. An HPLC chromatograph (LaChrome, Merck, Hitachi, Tokyo, JP) equipped, inter alia, with an autosampler, fluorescent detector, and thermostat oven columns (Merck), was used for the detection of vitamins. The analysis was performed on a Bionacom Velocity C18 PLMX column (4.6 × 250 mm, 5 µm), together with the precolumn (Bionacom LTD (London, UK). The measurement was made at the excitation and emission wavelengths of 360 and 503 nm, respectively, enabling the simultaneous determination of thiamine and riboflavin. The mobile phase used was a mixture of water (W) and acetonitrile (A) (acetonitrile for HPLC, POCH, Avantor Performance Materials Poland S.A., Gliwice, Poland). Gradient elution was performed as follows: *t* = 0, ratio 88:12 W/A; *t* = 12, ratio 0:100 W/A, temperature 22 °C. The external standards of thiamine (thiamine hydrochloride in HCl, purity ≥99.0% from Sigma-Aldrich, Merck KGaA, Darmstadt, Germany) and riboflavin (riboflavin in CH_3_CO_2_H, purity ≥98.0% from Sigma-Aldrich, Merck KGaA, Darmstadt, Germany) were used for the identification of these compounds and their quantitative analysis.

### 2.13. Data Analysis

The experiment was established using 3 repetitions per treatment, and the results are presented as the mean of 3 repetitions ± standard deviation (SD). Statistical analysis was performed with the Statistica 13.3 package (TIBCO Software Inc., Palo Alto, CA, USA). Differences between particular parameters were analyzed using one-way analysis of variance (ANOVA) and Tukey’s test, and *p*-values of less than or equal to 0.05 were considered to be statistically significant. The results were also examined for Pearson’s correlation coefficient (*r*) between analyzed parameters. Principal component analysis (PCA) was performed, and the first two components (PC1 and PC2) were used to derive biplots.

## 3. Results

### 3.1. Bioelements in Pleurotus Djamor Fruiting Bodies and Growing Substrate

The content of elements in the fruiting bodies of *P. djamor* was differentiated depending on the type and concentration of magnesium salts used in the experiment (Table 1). The highest content of magnesium was determined in mushrooms collected from substrate supplemented with the highest concentration of MgCl_2_ × 10 and MgSO_4_ × 10. These supplementation variants increased Mg content in fruiting bodies by 185% on average, compared to the control. The remaining supplementation treatments increased the content of Mg by 127% on average, and no significant differences were noted according to the salts used and their concentrations. Supplementation with MgCl_2_ × 0.5 resulted in the highest contents of potassium, iron, and copper in *P. djamor* fruiting bodies, whereas the lowest contents of these elements were found in samples from MgSO_4_ × 1 (K), MgSO_4_ × 10 (Fe), and MgSO_4_ × 0.5 (Cu) treatments. Moreover, K content was positively correlated with Fe (*r* = 0.542, *p ≤* 0.05) and Fe with Cu (*r* = 0.506, *p ≤* 0.05). The content of Ca and Zn was the highest in fruit bodies collected from control treatments. The antagonistic relation between Mg and Zn was confirmed by the correlation coefficient between these elements in *P. djamor* fruiting bodies (*r* = −0.537, *p ≤* 0.05). In general, the content of all elements was significantly lower in treatments supplemented with MgSO_4_ in comparison to MgCl_2_. The content of SO_4_^2−^ and Cl^−^ was higher in fruiting bodies supplemented with MgSO_4_ and MgCl_2_, respectively, as compared to the control. The correlation matrix for all the elements is demonstrated in subchapter 3.9. 

### 3.2. Biological Efficiency and Dry Weight Content

The highest biological efficiency was obtained for treatments supplemented with MgCl_2_ × 0.5 (73%) and the control (69%) (Figure 1). Fortification with the remaining concentrations of MgCl_2_ and MgSO_4_ decreased the biological efficiency of *P. djamor*; the lowest was noted for the maximum dose of Mg salts used in the experiment. The correlation coefficient between Mg content and biological efficiency was *r* = −0.769, *p ≤* 0.001 as it was demonstrated in subchapter 3.9. DW of fruiting bodies varied slightly but significantly between treatments (11.7–13.1 g 100 g^−1^ FW). The highest DW was determined in fruiting bodies collected from substrate supplemented with MgSO_4_ × 10, whereas the lowest related to the substrate supplemented with MgSO_4_ × 0.5. Fruit bodies from remaining treatments were not differentiated significantly concerning dry matter content. Contrary to biological efficiency, the dry weight of fruiting bodies was positively correlated with substrate supplementation with Mg (*r* = 0.547, *p ≤* 0.01) as it was demonstrated in Section 3.9.

### 3.3. Total Phenolics and DPPH^•^ Scavenging Activity

Total phenolic content and DPPH^•^ scavenging activity were statistically differentiated; the highest was in fruiting bodies collected from a substrate supplemented with MgCl_2_ × 10 (Figure 2), which was 50% higher than that of the control. Generally, supplementation with magnesium increased both parameters, with the exception of the MgSO_4_ × 0.5 and MgSO_4_ × 1 treatments for DPPH^•^. The stimulative effect of the Mg supplementation of the substrate on the phenolic content and DPPH^•^ scavenging activity in *P. djamor* fruiting bodies was supported by the coefficients of correlation between these parameters (*r* = 0.888, *p ≤* 0.001 for Mg and phenolics; *r* = 0.818, *p ≤* 0.001 for Mg and DPPH^•^ scavenging activity) as it was demonstrated in Section 3.9.

### 3.4. Soluble Sugar and L-Ascorbic Acid

Supplementation of the growing substrate with Mg chlorides and sulfides affected in different ways the soluble sugar content in *P. djamor* fruiting bodies. MgCl_2_ × 1 and MgCl_2_ × 10 caused a significant increase in soluble sugar in the investigated mushroom species, whereas MgSO_4_ × 1 application caused a decrease in soluble sugars, in contrast to MgCl_2_ × 10. This dose application resulted in a significant increase in soluble sugar content (Figure 3). L-ascorbic acid was the highest for fruit bodies sampled from the MgCl_2_ × 10 treatment. Moreover, the determined value was 48% higher than that of other samples. Lower MgCl_2_ doses did not significantly affect L-ascorbic acid content in *P. djamor*, similarly to the MgSO_4_ × 0.5 treatment. Higher MgSO_4_ doses increased L-ascorbic acid content in fruiting bodies of *P. djamor*. In general, a positive correlation was noted for Mg and L-ascorbic acid content (*r* = 0.817, *p ≤* 0.001) as it was demonstrated in Section 3.9.

### 3.5. Thiamine and Riboflavin

The supplementation of *P. djamor* substrate with MgSO_4_ at all doses significantly increased the thiamine content in fruiting bodies (by 21% on average for the doses) and riboflavin (with the exception of MgSO_4_ × 1) (Figure 4). Moreover, thiamine content was positively correlated with Mg (*r* = 0.545, *p ≤* 0.05), but negatively correlated with the other elements, including Ca (*r* = −0.786, *p ≤* 0.001), Fe (*r* = −0.793, *p* ≤ 0.001), Zn (*p* = −0.548, *p* ≤ 0.001), and Cu (*r* = −0.604, *p* ≤ 0.01) as it was demonstrated in subchapter 3.9. MgCl_2_ application to the growing substrate had no significant effect on the thiamine content in fruiting bodies, but caused a decrease in riboflavin, when Mg salts were applied in doses of MgCl_2_ × 0.5 and MgCl_2_ × 10. Riboflavin content was negatively correlated with K (*r* = −0.621, *p* ≤ 0.01), Ca (*p* = −0.753, *p* ≤ 0.001), Fe (*r* = −0.507, *p* ≤ 0.01), and Cu (*r* = −0.712, *p* ≤ 0.001). Both vitamins of group B ware negatively correlated with glutathione content in *P. djamor* fruiting bodies (*r* = −0.627, *p* ≤ 0.05, and *r* = −0.442, *p* ≤ 0.05, respectively) as it was demonstrated in Section 3.9.

### 3.6. p-Hydroxybenzoic Acid and Phenylalanine

The application of MgSO_4_ salts to the growing substrate caused an increase in *p*-hydroxybenzoic acid in *P. djamor* fruiting bodies, from 9% (MgSO_4_ × 10) to 46% (MgSO_4_ × 1) (Figure 5). Concerning the supplementation with chlorides, a significant effect of *p*-hydroxybenzoic acid in fruiting bodies was observed only for MgCl_2_ × 0.5 treatment. Substrate supplementation with MgCl_2_ × 0.5 and MgSO_4_ × 0.5 and MgSO_4_ × 1 significantly increased phenylalanine content in *P. djamor* fruiting bodies by eight, ten-, and nine-fold, respectively, relative to the control.

### 3.7. L-Tryptophan and 5-Hydroxy-L-tryptophan

The supplementation of the substrate with both chlorides and sulfates of Mg caused a significant increase in L-tryptophan content in *P. djamor* fruiting bodies. A similar observation was noted for 5-hydroxy-L-tryptophan, with the exception of the MgCl_2_ × 1 treatment. L-tryptophan was positively correlated with Mg content (*r* = 0.643, *p* ≤ 0.05) (Figure 6) as it was demonstrated in Section 3.9.

### 3.8. Glutathione and Lovastatin

The glutathione content was the highest in the control. Generally, supplementation decreased glutathione content, with the lowest value noted for the treatment MgSO_4_ × 1 (Figure 7). Generally, supplementation with MgCl_2_ reduced glutathione content to a lower extent (by 15%, average for doses), whereas MgSO_4_ increased it to a higher extent (by 46%, average for doses), compared to the control. It is interesting that glutathione content was positively correlated with crucial mineral compounds, including K (*r* = 0.680), Ca (*r* = 0.751), Fe (*r* = 0.826), and Zn (*r* = 0.688), with *p* ≤ 0.001 in all cases (Figure 8). Lovastatin content was differentiated without any regular trend for experimental treatments. Only substrate supplementation with MgSO_4_ × 0.5 significantly increased the content of this compound in *P. djamor* fruiting bodies by 48% compared to the control.

### 3.9. Correlation Matrix and Bi-Plot Presentation of Correlations

PCA illustrated that MgSO_4_ × 0.5 and MgSO_4_ × 1 treatments contributed significantly and negatively to PC 1, whereas MgSO_4_ × 10 contributed significantly and negatively to PC 2. MgCl_2_ × 10 treatment contributed significantly and negatively to PC 2 but positively to PC 1. MgCl_2_ × 0.5 contributed significantly and positively to PC 2 (Figure 9). Based on the PCA analysis (Figure 10), magnesium results were in contrast to those of biological efficiency and most of the other bioelements analyzed (Figure 5a,b). Concerning organic compounds, magnesium was located in a separate group, consisting of parameters responsible for antioxidant activity, as total phenols, L-ascorbic acid, and L-tryptophan.

## 4. Discussion

### 4.1. Bioelements in P. djamor Fruiting Bodies and Growing Substrate

The substrate supplementation with magnesium caused a significant increase of its concentration in the fruiting bodies of *P. djamor*; both Mg salts, MgSO_4_ and MgCl_2_, showed similar effectiveness when applied in the same doses. The remaining bioelement content was significantly lower in fruiting bodies collected from treatments supplemented with MgSO_4_ in comparison to MgCl_2_. The percentage of the dietary value (% DV) is used to define a food serving as high or low in an individual nutrient. In general, 5% DV or less of a nutrient per serving is considered low, whereas 20% DV or more of a nutrient per serving is considered high. The allowance for Mg from the 2015–2020 Dietary Guidelines for Americans is equal to 420 mg per day per adult [32]. A quantity of 100 g of fresh *P. djamor* fruiting bodies supplemented with Mg salts in the present research contained from 27% DV (MgCl_2_ × 1) to 41% DV (MgCl_2_ × 10); thus, supplementation can be an effective technique to produce mushrooms rich in Mg as a dietary component. Although *P. djamor* grown in supplemented substrate showed lower productivity, this was only evident in the fresh weight. Differences in dry weight were negligible and, in most treatments, comparable with the control.

In the present research fruiting bodies of *P. djamor*, not supplemented with Mg, contained Mg—738, K—658, Ca—27, Fe—6.6, Zn—0.1, and Cu—0.1 mg 100 g^−1^ DW. The analysis of the concentration of these elements showed both similarities and differences compared to the data from the literature [43,44]. For example, in the recent research of Mleczek et al. [25] the most abundant minerals found in *P. djamor* fruiting bodies were K 2000–2390, Ca 108–191, Mg 52–71, Fe 0.2–0.6, Zn 5,8, and Cu 1,3 mg 100 g^−1^ DW, depending on the substrate composition. Siwulski et al. [45] classified minerals detected in mushroom fruiting bodies into five groups; three of these comprising the most abundant elements are as follows: (i) exceeding 1000 mg kg^−1^ DW (K, P, Mg, and Ca); (ii) ranging from 100 to 1000 mg kg^−1^ (Fe, Na, Zn); (iii) ranging from 10 to 100 mg kg^−1^ (Cu, Al, Mn, B). Potassium and magnesium were reported as the most abundant elements in *Pleurotus* species and many other edible mushrooms, both cultivated and collected from natural ecosystems [46,47]. Ca, Fe, Zn, and Cu content in *P. djamor* fruiting bodies in the present research was lower than values classified by Siwulski et al. [45], and Krakowska et al. [5], but comparable to the values mentioned by Vieira et al. [48] for *P. ostreatus*. Interestingly, magnesium in the fruiting bodies of *P. djamor* was determined at a higher level than can be found in the literature. Considering that mushrooms’ mineral composition is highly related to the substrate, growing conditions, flux, etc., supplementation of fruiting bodies with lower initial magnesium content can be more effective. The present research showed that *P. djamor* supplementation with Mg affected the fruit body’s composition concerning the other macro- and micronutrients. The most significant was the antagonism between Mg and Zn, which was confirmed with a negative correlation coefficient.

### 4.2. Biological Efficiency and Dry Weight Content

The biological efficiency of the specific substrates is an essential factor that decides on their suitability to cultivate a particular species or strain of mushrooms. The substrates can be processed either by composition or pasteurization, and further additions that affect yield quality and quantity [49,50]. The supplementation of *Pleurotus* spp. has not always resulted in a higher yield of fruit bodies [25]. However, the main target of modern mushroom production is not always accelerated biological efficiency, but rather a high level of bioactive compounds, which determine the market quality of the raw material. Moreover, the decrease in biological efficiency of *P. djamor* following Mg supplementation was caused by the decrease in water content, with dry matter yield slightly differentiated between experimental treatments. It appears that the biological efficiency and dry weight production is related to the salt used as a supplement. In selenium-fortified oyster mushrooms, de Oliveira and Naozuka [27] did not notice alterations in the moisture of fruiting bodies with the increase in Se concentration in the culture medium. Furthermore, the biological efficiency showed that Se enrichment did not alter the potential of the fungus to biodegrade the organic substrate.

### 4.3. Organic Compounds and Antioxidant Activity

Fungal major bioactive compounds, known as mycochemicals, are naturally found in the *Pleurotus* spp. fruiting bodies and their concentration may be increased by modifying the substrate composition, culture, or postharvest conditions [51]. Some of these are phenolic compounds that could be extracted and included in formulations to prevent oxidative stress [52]. Generally, supplementation with Mg in the present research increased the phenolic content and DPPH^•^ scavenging activity, with the exception of two treatments with lower MgSO_4_ content for DPPH^•^. Moreover, the stimulative effect of Mg supplementation of substrate on *P. djamor* fruiting bodies was supported by close eigenvectors of PC analysis and a positive correlation coefficient between these parameters, which was also reported by Puttaraju et al. [53]. By contrast, Vieira et al. [48] demonstrated that supplementation with iron, zinc or lithium reduced antioxidant activity in *P. ostreatus* fruiting bodies because polyphenol groups formed complexes with metal ions such as Fe and Zn, manifesting a reduced availability for free radicals’ donation and lower antioxidant activity. Thus, it appears clear that the antagonism, which in the present research was statistically significant for Mg and Zn, and slightly notable for Mg and Fe, affects the manifestation of polyphenols’ antioxidant activity. However, it cannot be ruled out that the total antioxidant activity may also relate to other compounds present in *P. djamor* fruiting bodies, such as L-tryptophan and L-ascorbic acid.

Carbohydrates in mushrooms are involved in structural composition, but are essential in maintaining the high osmotic concentration and providing the source of energy. Due to their wide range of celluloid substances, including dietary fiber, mushrooms can be part of a low-calorie diet with higher therapeutic value [18]. Glucose, mannitol, and trehalose are abundant sugars in cultivated edible mushrooms, but fructose and sucrose are found in low amounts [54]. Soluble sugars can contribute to positive health characteristics of *P. djamor* fruiting bodies. Recently, Maity et al. [55] isolated from pink oyster mushroom fruiting bodies a soluble galactoglucan of moderate DPPH^•^ scavenging activity that increased in a dose-dependent manner. The authors suggested that the isolated compound can be used as natural antioxidant. The positive correlation between soluble sugars determined in the present research contributes to the general definition of mushroom carbohydrates as a biologically active molecules that can be active components in functional products due to their antioxidant advantages.

Mushrooms are also a good source of vitamins, with high riboflavin, niacin, folates, and traces of vitamin C, B_1_, B_12_, D, and E [54]. One hundred grams of fresh *P. ostreatus* fruiting bodies provides 15% of the recommended daily intake of L-ascorbic acid, and 40% of niacin, riboflavin, and thiamin [56]. Supplementation of *P. djamor* substrate with magnesium sulfate in all doses significantly increased thiamine content in fruiting bodies, and riboflavin in most cases; thiamine content was positively correlated with Mg, confirming the effectiveness of Mg supplementation concerning thiamine content. MgCl_2_ application to the growing substrate had no significant effect on thiamine content in fruiting bodies but caused the decrease in riboflavin. The thiamine and riboflavin contents determined by Goyal et al. [57] for *P. sajor-caju* were 4.13 and 3.71 mg 100 g^−1^ DW, respectively. By comparison, in the present study, the values were 0.48 and 0.38 mg 100 g^−1^ DW for thiamine, and 2.1 and 1.6 mg 100 g^−1^ DW for riboflavin, in *P. djamor* fruiting bodies supplemented with MgSO_4_ and MgCl_2_, respectively. On a fresh weight basis, L-ascorbic acid content was 4.34 mg 100 g^−1^ in *P. sajor-caju* mushrooms in the study of Goyal et al. [57], whereas *P. djamor* contained 43 and 53 mg 100 g^−1^ FW, depending on substrate supplementation with Mg sulfides or chlorides, respectively. Based on the present results, pink oyster mushrooms can significantly contribute to functional food composition as an excellent source of the aforementioned vitamins.

*Pleurotus* spp. are a good source of proteins, comprising all of the essential amino acids with excellent digestibility. Non-protein nitrogen compounds include amino acids, chitin, and nucleic acids. Some amino acids contribute to the taste of mushrooms, which is highly valued by consumers [58]. Phenylalanine, L-tryptophan, and 5-hydroxy-l-tryptophan, determined in the *P. djamor* fruiting bodies in the present research, are essential aromatic amino acids, and act as precursors for neurotransmitters, such as serotonin, and catecholamines. L-tryptophan is the precursor of vitamin B_3_, which is a stimulator of insulin secretion and growth hormone [59]. The present research proved the possibility of linking Mg supplementation with an increased L-tryptophan level in *P. djamor* fruiting bodies. Some experimental treatments also increased the level of p-hydroxybenzoic acid, 5-hydroxy-l-tryptophan, and phenylalanine. However, the correlation between magnesium and L-tryptophan was the most notable, and was confirmed statistically with a significant and positive correlation coefficient. L-Tryptophan also contributed significantly to the total antioxidant activity of *P. djamor* fruiting bodies supplemented with Mg. The biological role of L-tryptophan in free radical scavenging in *Pleurotus* spp. was reported by Jegadeesh et al. [1]. Consumption of *P. ostreatus* may contribute to the cysteine pool. The cysteine which is a precursor in glutathione synthesis can affect the functions of glutathione [60]. In the present research, the glutathione content was the highest in the control and, generally, supplementation decreased the content of glutathione.

Lovastatin content was differentiated without any regular trend for experimental treatments and varied from 0.20 to 10.5 mg 100 g^−1^ DW. Only substrate supplementation with the lowest dose significantly increased lovastatin content in *P. djamor* fruiting bodies. Lovastatin is characterized by a high degree of differentiation in its content in mushrooms, depending on the species, location, growing conditions, technologies of cultivation, etc. [56]. For example, Krakowska et al. [5] determined 7.76, 1.18, 1.14, and 0.39 mg 100 g^−1^ DW of lovastatin in the fruiting bodies of *P. citrinopileatus, P. florida, P. ostreatus*, and *P. eryngi*, respectively.

## 5. Conclusions

The cultivation of pink oyster mushrooms in a medium enriched with magnesium salts proved to be an effective technique for producing fruiting bodies with a chemical profile that allows them to be classified as a functional food. Mg sulfates and chlorides applied to the substrate to grow *P. djamor* effectively increased Mg content in fruiting bodies and significantly altered the content of all analyzed elements, albeit in different ways. Based on the results of the present experiment, and the cited literature, significant differentiation in the mineral composition of mushroom fruiting bodies was confirmed. Mg supplementation was proven to be effective in the increase in organic compounds that contribute to the antioxidant activity of pink oyster mushrooms. Based on these results, the supplementation of a substrate with major and trace elements, which is crucial for the utilization of mushrooms as a functional food, appears to be a promising technique with which to forecast and standardize the chemical composition of fruiting bodies.

## Figures and Tables

**Figure 1 molecules-26-03273-f001:**
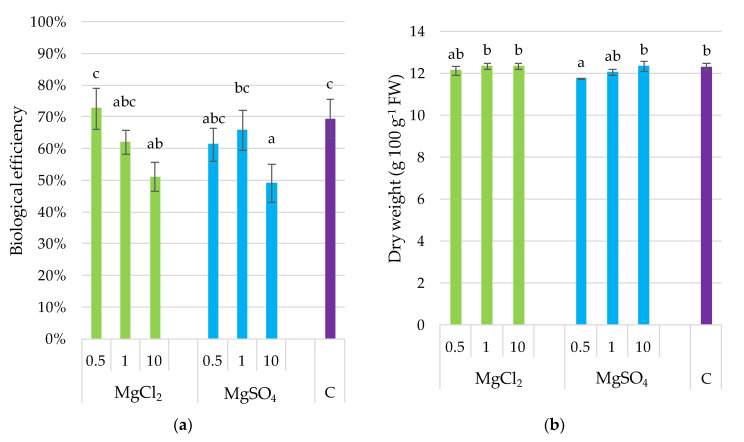
Biological efficiency (**a**) and dry weight (**b**) content in *P. djamor* fruiting bodies depending on Mg salts supplementation, Bars marked with different letters (a,b,c) are significantly different *p* ≤ 0.05 according to Tukey’s test N = 6. MgCl_2_ × 0.5—210 mg of Mg; MgCl_2_ × 1—420 mg of Mg; MgCl_2_ × 10—4200 mg of Mg (in the form of MgCl_2_ per 2 kg of substrate); MgSO_4_ × 0.5—210 mg of Mg; MgSO_4_ × 1—420 mg of Mg; MgSO_4_ × 10—4200 mg (in the form of MgSO_4_ per 2 kg of substrate); C—standard growing medium.

**Figure 2 molecules-26-03273-f002:**
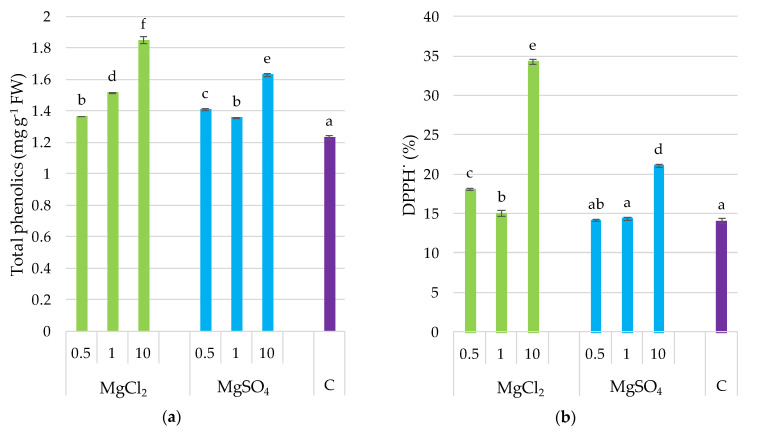
Total phenolics content and DPPH^•^ scavenging activity in *P. djamor* fruiting bodies depending on Mg salts supplementation. (**a**) Total phenolics content; (**b**) DPPH^•^ scavenging activity. Each bar represents mean value ± standard deviation. Bars marked with different letters (a,b,c,d,e,f) are significantly different *p ≤* 0.05 according to Tukey’s test, N = 6. MgCl_2_ × 0.5—210 mg of Mg; MgCl_2_ × 1—420 mg of Mg; MgCl_2_ × 10—4200 mg of Mg (in the form of MgCl_2_ per 2 kg of substrate); MgSO_4_ × 0.5—210 mg of Mg; MgSO_4_ × 1—420 mg of Mg; MgSO_4_ × 10—4200 mg (in the form of MgSO_4_ per 2 kg of substrate); C—standard growing medium.

**Figure 3 molecules-26-03273-f003:**
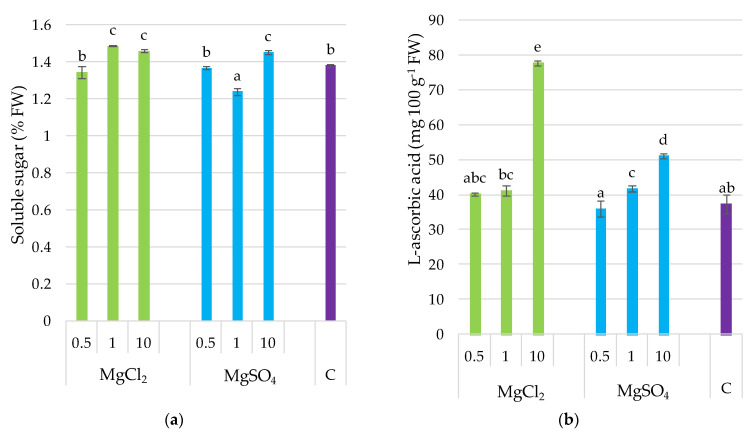
Soluble sugar and L-ascorbic acid content in *P. djamor* fruiting bodies depending on Mg salts supplementation. (**a**) Soluble sugar content; (**b**) L-ascorbic acid content. Each bar represents mean value ± standard deviation. Bars marked with different letters (a,b,c,d,e) are significantly different *p ≤* 0.05 according to Tukey’s test, N = 6. MgCl_2_ × 0.5—210 mg of Mg; MgCl_2_ × 1—420 mg of Mg; MgCl_2_ × 10—4200 mg of Mg (in the form of MgCl_2_ per 2 kg of substrate); MgSO_4_ × 0.5—210 mg of Mg; MgSO_4_ × 1—420 mg of Mg; MgSO_4_ × 10—4200 mg (in the form of MgSO_4_ per 2 kg of substrate); C—standard growing medium.

**Figure 4 molecules-26-03273-f004:**
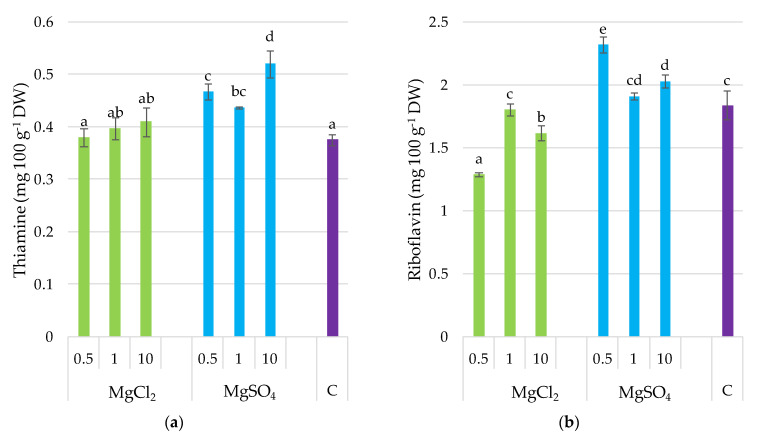
Thiamine and riboflavin content in *P. djamor* fruiting bodies depending on Mg salts supplementation. (**a**) Thiamine content; (**b**) Riboflavin content. Each bar represents mean value ± standard deviation. Bars marked with different letters (a,b,c,d,e) are significantly different *p* ≤ 0.05 according to Tukey’s test, N = 6. MgCl_2_ × 0.5—210 mg of Mg; MgCl_2_ × 1—420 mg of Mg; MgCl_2_ × 10—4200 mg of Mg (in the form of MgCl_2_ per 2 kg of substrate); MgSO_4_ × 0.5—210 mg of Mg; MgSO_4_ × 1—420 mg of Mg; MgSO_4_ × 10—4200 mg (in the form of MgSO_4_ per 2 kg of substrate); C—standard growing medium.

**Figure 5 molecules-26-03273-f005:**
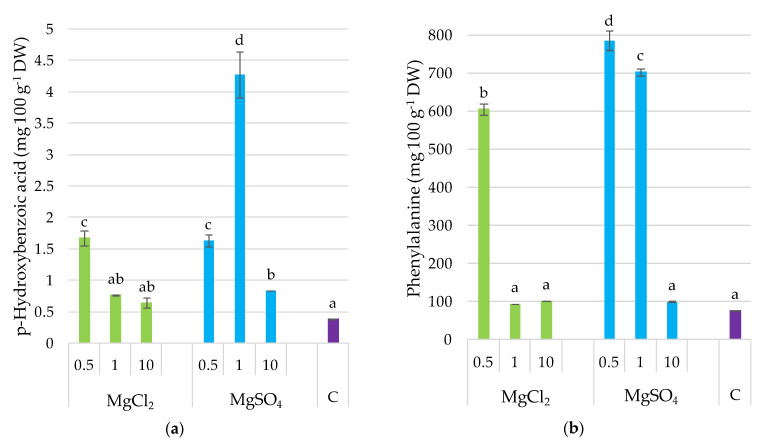
*p*-Hydroxybenzoic acid and phenylalanine content in *P. djamor* fruiting bodies depending on Mg salts supplementation. (**a**) *p*-Hydroxybenzoic acid content; (**b**) Phenylalanine content. Each bar represents mean value ± standard deviation. Bars marked with different letters (a,b,c,d) are significantly different *p* ≤ 0.05 according to Tukey’s test, N = 6. MgCl_2_ × 0.5—210 mg of Mg; MgCl_2_ × 1—420 mg of Mg; MgCl_2_ × 10—4200 mg of Mg (in the form of MgCl_2_ per 2 kg of substrate); MgSO_4_ × 0.5—210 mg of Mg; MgSO_4_ × 1—420 mg of Mg; MgSO_4_ × 10—4200 mg (in the form of MgSO_4_ per 2 kg of substrate); C—standard growing medium.

**Figure 6 molecules-26-03273-f006:**
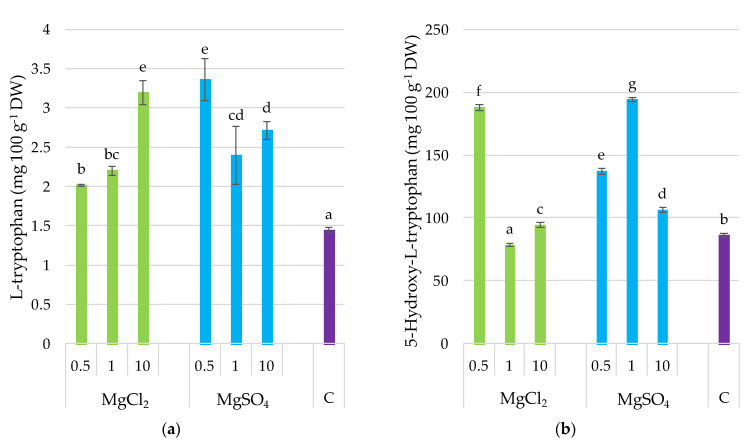
L-Tryptophan and 5-hydroxy-l-tryptophan content in *P. djamor* fruiting bodies depending on Mg salts supplementation. (**a**) L-Tryptophan content; (**b**) 5-hydroxy-l-tryptophan content. Each bar represents mean value ± standard deviation. Bars marked with different letters (a,b,c,d,e,f,g) are significantly different *p* ≤ 0.05 according to Tukey’s test, N = 6. MgCl_2_ × 0.5—210 mg of Mg; MgCl_2_ × 1—420 mg of Mg; MgCl_2_ × 10—4200 mg of Mg (in the form of MgCl_2_ per 2 kg of substrate); MgSO_4_ × 0.5—210 mg of Mg; MgSO_4_ × 1—420 mg of Mg; MgSO_4_ × 10—4200 mg (in the form of MgSO_4_ per 2 kg of substrate); C—standard growing medium.

**Figure 7 molecules-26-03273-f007:**
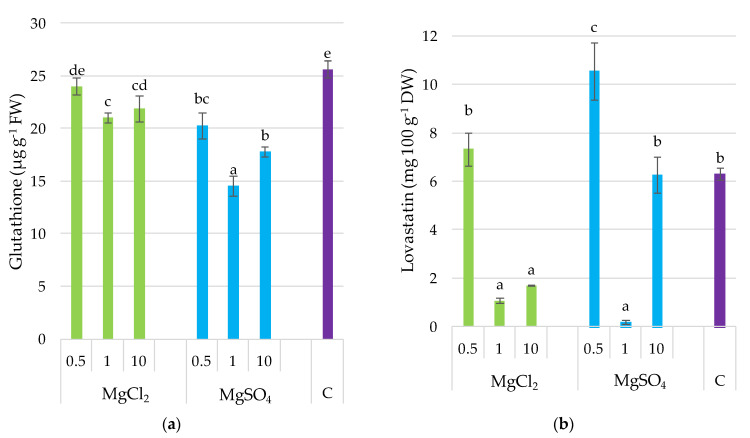
Glutathione and lovastatin content in *P. djamor* fruiting bodies depending on Mg salts supplementation. (**a**) Glutathione content; (**b**) Lovastatin content. Each bar represents mean value ± standard deviation. Bars marked with different letters (a,b,c,d,e) are significantly different *p* ≤ 0.05 according to Tukey’s test, N = 6. MgCl_2_ × 0.5—210 mg of Mg; MgCl_2_ × 1—420 mg of Mg; MgCl_2_ × 10—4200 mg of Mg (in the form of MgCl_2_ per 2 kg of substrate); MgSO_4_ × 0.5—210 mg of Mg; MgSO_4_ × 1—420 mg of Mg; MgSO_4_ × 10—4200 mg (in the form of MgSO_4_ per 2 kg of substrate); C—standard growing medium.

**Figure 8 molecules-26-03273-f008:**
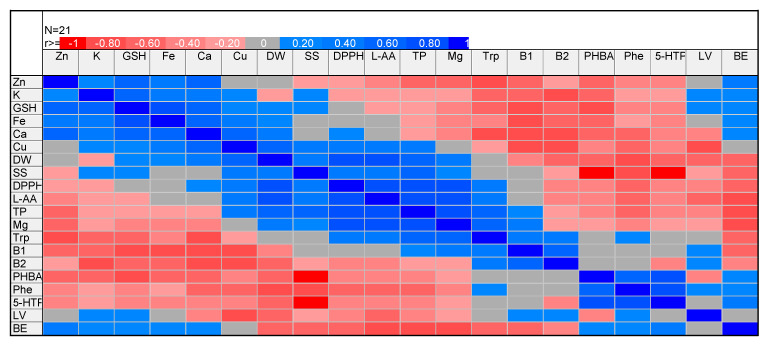
Correlation matrix of *P. djamor* productivity and quality parameters. Abbreviations: Zn—zinc, K—potassium, GSH—glutathione, Fe—iron, Ca—calcium, Cu—copper, DW—dry weight, SS—soluble sugar, DPPH—DPPH^•^ scavenging activity, L-AA—L-ascorbic acid, TP—total phenolics, Mg—magnesium, Trp—L-tryptophan, B_1_—thiamine, B_2_—riboflavin, PHBA—*p*-hydroxybenzoic acid Phe—phenylalanine, 5-HTP—5-hydroxy-l-tryptophan, LV—lovastatin, BE—biological efficiency.

**Figure 9 molecules-26-03273-f009:**
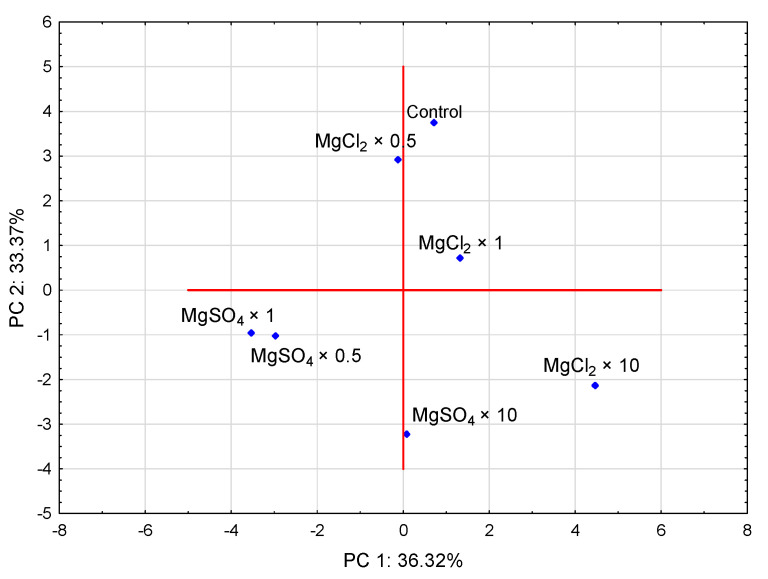
Bi-plot presenting the correlation between the tested treatments. MgCl_2_ × 0.5—210 mg of Mg, MgCl_2_ × 1—420 mg of Mg, MgCl_2_ × 10—4200 mg of Mg (in a form of MgCl_2_ per 2 kg of substrate); MgSO_4_ × 0.5—210 mg of Mg, MgSO_4_ × 1—420 mg of Mg, MgSO_4_ × 10—4200 mg (in a form of MgSO_4_ per 2 kg of substrate).

**Figure 10 molecules-26-03273-f010:**
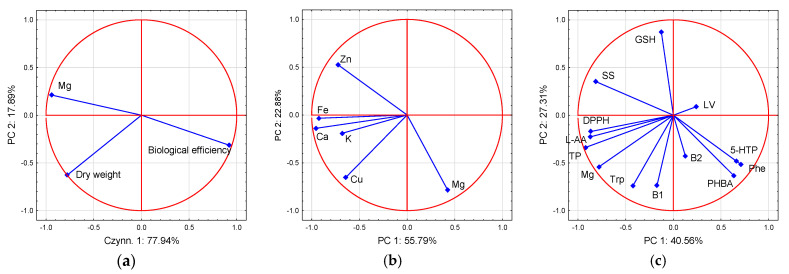
Bi-plot presenting the correlation between the tested magnesium (Mg) and yield quantity (**a**); macro- and micronutrients (**b**), and organic compounds and antioxidant efficiency (**c**). Abbreviations: Zn—zinc, K—potassium, GSH—glutathione, Fe—iron, Ca—calcium, Cu—copper, DPPH—DPPH^•^ scavenging activity, 5-HTP—5-hydroxy-l-tryptophan, B_1_—thiamine, B_2_—riboflavin, L-AA—L-ascorbic acid, LV—lovastatin, PHBA—*p*-hydroxybenzoic acid, Phe—phenylalanine, SS—soluble sugar, TP—total phenolics, Trp—L-tryptophan.

**Table 1 molecules-26-03273-t001:** The content of elements and anions in fruiting bodies and growing substrate of *P. djamor* depending on magnesium salts supplementation (mg 100 g^−1^ DW).

Treatment	Mg	K	Ca	Fe	Zn	Cu	Cl^−^	SO_4_^2^^−^
Control	783 ± 28 ^a1^	658 ± 6 ^b^	27.2 ± 2.9 ^c^	6.63 ± 0.44 ^c^	43.3 ± 7.2 ^b^	4.03 ± 0.25 ^c^	81 ± 11 ^a^	428 ± 25 ^a^
MgCl_2_ × 0.5 ^2^	1015 ± 23 ^b^	759 ± 20 ^c^	25.1 ± 0.9 ^bc^	6.52 ± 0.81 ^c^	18.6 ± 3.1 ^a^	4.69 ± 0.03 ^d^	192 ± 10 ^b^	n.a. ^3^
MgCl_2_ × 1	934 ± 81 ^b^	682 ± 29 ^bc^	20.9 ± 0.8 ^b^	5.95 ± 0.01 ^bc^	15.2 ± 1.5 ^a^	5.56 ± 0.02 ^e^	275 ±13 ^c^	n.a.
MgCl_2_ × 10	1455 ± 22 ^c^	624 ± 28 ^ab^	24.3 ± 1.9 ^bc^	6.42 ± 0.69 ^c^	17.1 ± 2.2 ^a^	5.99 ± 0.26 ^e^	351 ± 11 ^d^	n.a.
MgSO_4_ × 0.5	1008 ± 25 ^b^	614 ± 23 ^ab^	11.9 ± 1.3 ^a^	5.71 ± 0.07 ^abc^	13.9 ± 0.1 ^a^	2.28 ± 0.28 ^a^	n.a.	694 ± 25 ^b^
MgSO_4_ × 1	1014 ± 15 ^b^	564 ± 55 ^a^	15.6 ± 1.4 ^a^	4.73 ± 0.21 ^ab^	12.2 ± 0.9 ^a^	3.51 ± 0.26 ^bc^	n.a.	765 ± 12 ^b^
MgSO_4_ × 10	1434 ± 8 ^c^	640 ± 9 ^ab^	13.4 ± 0.9 ^a^	4.64 ± 0.14 ^a^	11.3 ± 0.9 ^a^	3.05 ± 0.31 ^b^	n.a.	1043 ± 76 ^c^
Growing substrate	921 ± 23	47 ± 3	37.7 ± 2.6	1.77 ± 0.21	3.5 ± 0.6	4.25 ± 0.31	64 ± 9	* ^4^

^1^ Means in a column followed by different superscript letters (a,b,c,d,e) are significantly different *p* ≤ 0.05 according to Tukey’s test, N = 6. Each value represents the mean ± standard deviation. ^2^ MgCl_2_ × 0.5—210 mg of Mg, MgCl_2_ × 1—420 mg of Mg, MgCl_2_ × 10—4200 mg of Mg (in a form of MgCl_2_ per 2 kg of substrate); MgSO_4_ × 0.5—210 mg of Mg, MgSO_4_ × 1—420 mg of Mg, MgSO_4_ × 10—4200 mg (in a form of MgSO_4_ per 2 kg of substrate). ^3^ n.a. = Not analyzed in a sample. ^4^ * = Below the detection level.

## Data Availability

Data is contained within the article.

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
