# Peer review of "Supplementation with Magnesium Salts—A Strategy to Increase Nutraceutical Value of Pleurotus djamor Fruiting Bodies"

_molecules, 2021, doi:10.3390/molecules26113273_

Round 1
Reviewer 1 Report
The manuscript entitled "Supplementation with magnesium salts – the strategy to increase nutraceutical value of Pleurotus djamor fruiting bodies" contains new and significant information in the field.
It demonstrates an adequate understanding of the relevant literature in the field and cite an appropriate range of literature sources. The paper's argument is built on an appropriate base of theory, concepts and other ideas.
However, there were some minor points that need to be adressed before publication. Therefore I recommend the manuscript to be published after negligible revision.
The paper is not identify clearly implications for research, practice and society, in that sense I suggest you to provide some information, this will increase the manuscript value.
Also, I kindly recommend the next paper to be consulting for the introduction section of the manuscript: DOI: http://dx.doi.org/10.15835/buasvmcn-fst:12629
I would like to congratulate the authors for the quality and soundness of data presented.
Author Response
Reviewer 1
The manuscript entitled "Supplementation with magnesium salts – the strategy to increase nutraceutical value of Pleurotus djamor fruiting bodies" contains new and significant information in the field.
It demonstrates an adequate understanding of the relevant literature in the field and cite an appropriate range of literature sources. The paper's argument is built on an appropriate base of theory, concepts and other ideas.
However, there were some minor points that need to be adressed before publication. Therefore I recommend the manuscript to be published after negligible revision.
Author response: Thank you for you the correction of our manuscript, for accurate comments, and positive opinion. We corrected the paper according to all your valuable remarks.
The paper is not identify clearly implications for research, practice and society, in that sense I suggest you to provide some information, this will increase the manuscript value. Also, I kindly recommend the next paper to be consulting for the introduction section of the manuscript: DOI: http://dx.doi.org/10.15835/buasvmcn-fst:12629
Author response: We supplemented our MS with proposed literature and we added information about basic chemical compositions of Pleurotus ostreatus collected in wild habitats. We also added some ideas on the implications of the present studies for research, practice, and society.
I would like to congratulate the authors for the quality and soundness of data presented.
Author response: Thank you for your positive opinion about our MS.
Reviewer 2 Report
In the present manuscript, the effect of substrate supplementation with magnesium (Mg) chloride and sulfide salts on Mg content, biomass, and chemical composition of pink oyster mushroom (Pleurotus djamor) fruiting bodies was evaluated.
Give more details about the experimental design, the number of bags and cubes per treatment. -
Line : the authors mention 3 flushes for sample harvesting. Do they mean three different harvestings or three batch samples?
The treatments have to better described to avoid any confusions. The authors mention three concentrations in Lines 109-113 but it is not clear whether they mean two different treatments for each concentration as it is presented in the results section.
I suggest the biological efficiency to be expressed taking into account the amount of Mg provided and the Mg absorbed by the mycelium.
Figure 1b: check statistics because the C treatment does not seem to differ from MgSO4 10 treatment. The same applies for Figure 3b.
Figure 6a: there should be a treatment with (d) denotation. Check the statistics.
Lines 499-500 and 532-535: this argument is not true since no significant difference were recorded in dry weight of MgCl2 treatments and the control treatment.
Author Response
Reviewer 2
Comments and Suggestions for Authors
In the present manuscript, the effect of substrate supplementation with magnesium (Mg) chloride and sulfide salts on Mg content, biomass, and chemical composition of pink oyster mushroom (Pleurotus djamor) fruiting bodies was evaluated.
Author response: Thank you for the comprehensive review of our MS.
Give more details about the experimental design, the number of bags and cubes per treatment.
Author response: We added this information in methodology (2.2 chapter).
Line : the authors mention 3 flushes for sample harvesting. Do they mean three different harvestings or three batch samples?
Author response: We corrected methodology (2.2 chapter) and we described the experimental layout more precisely, addressing this comment.
The treatments have to better described to avoid any confusions. The authors mention three concentrations in Lines 109-113 but it is not clear whether they mean two different treatments for each concentration as it is presented in the results section.
Author response: We changed methodology (2.1 chapter) and we described the experimental layout more precisely, addressing this comment.
I suggest the biological efficiency to be expressed taking into account the amount of Mg provided and the Mg absorbed by the mycelium.
Author response: We are not able to provide the amount Mg absorbed by the mycelium because we didn’t not analyze Mg content in mycelium grown inside the substrate. We analyzed only fruiting bodies as a source of raw material of food or pharmacological usage. We recognize that absorbed Mg was divided between mycelium and fruiting bodies. This can be an important issue for future investigation. In our previous studies we compared Zn and Se accumulation in fruiting bodies from standard cultivation and mycelium from in vitro cultures https://doi.org/10.3390/molecules25040889 . The problem with analyzing the mycelium from the standard cultivation is lack of technical methods to isolated it from substrate residues.
Figure 1b: check statistics because the C treatment does not seem to differ from MgSO4 10 treatment. The same applies for Figure 3b.
Author response: We have checked our statistic and we corrected our error with indication of homogenous groups.
Figure 6a: there should be a treatment with (d) denotation. Check the statistics.
Author response: Thank you for an indication of this error, we corrected it.
Lines 499-500 and 532-535: this argument is not true since no significant difference were recorded in dry weight of MgCl2 treatments and the control treatment.
Author response: Oyster mushrooms are usually consumed as fresh fruiting bodies. Presented data (L499-500), refer to daily consumption of fresh weight of oyster mushroom and (L532-535) high level of bioactive compounds in a raw material. According to Dietary Guidelines for Americans, products that contain more than 20% of Mg can considered diet supplements. Although there was no significant difference were in dry weight of MgCl2 treatments and the control treatment, we confirmed significant differences in Mg content expressed in DW. We recounted it to fresh weight to evaluate DV.
Reviewer 3 Report
The discussion and conclusion was elaborate and the manuscript can be considered for publication. However, the entire manuscript requires improvement in English language and the authors should revise the manuscript according to the comments listed below. The study appeared to have no true replication of the treatment – according to Section 2.12 – each sample was analysed in three independent repetitions! Please clarify.
Abstract: Overall it is difficult to grasp when in the production stage of the mushroom that the supplementation with Mg occurred
Line 24: Please specify % decrease in productivity
Line 25: Please specify the lowest concentration of MgCl2 causing a reduction in productivity
Lines 25-26: This sentence is not clear “but the effect on dry matter content in fruiting bodies was slight but significant”
Line 26: Please specify % increase in Mg content of the mushroom
Lines 26-27: This sentence is not clear “modified in a differentiated way”
What does it mean by “pro health effects”
Lines 56 and 58: Remove the dash/hyphen for “mycelium-had” and “mice – this”
Lines 65-66: This sentence is not clear “Although this species is less important than grey oyster mushroom (P. ostreatus) considering the economical approach”
Lines 72-75 is too long and difficult to understand
Please clarify the term “slat-dependent effect”
Line 109: This sentence is not clear “Mentioned magnesium salts we applied”
Line 119: Please specify the model of equipment used to perform the sterilisation
Line 127: “1.5 h” repeated twice in the same sentence
Line 130: Please specify the model of the cultivation chamber
Line 154: Please clarify “reference solution”
Section 2.6: Since the sample is pink oyster mushroom, was the extract pink in colour – which could have interfere with the vitamin C assay?
Line 225: Please clarify the term “ samples from intestinal juice”
Section 2.10.1: Please clarify the DAD detection of phenylalanine and phenolics acids were done at which wavelength. Also, please specify the purify of the standards used for quantification.
Section 2.10.2: Please specify the purity of the standards used for quantification
Section 2.10.3: The compound detection and quantification is not clearly described here
Section 2.11: The purity of thiamine standard needs to be specified
Section 2.12: The authors present correlation result in the manuscript – please clarify which correlation test was performed here.
Table 1 – What is the “detection level” for sulphate?
Table 1 – It is not clear why samples supplemented with MgSO2 not analysed for chloride
It is not clearly described in the manuscript what is the “standard growing condition”
Author Response
Reviewer 3
The discussion and conclusion was elaborate and the manuscript can be considered for publication. However, the entire manuscript requires improvement in English language and the authors should revise the manuscript according to the comments listed below. The study appeared to have no true replication of the treatment – according to Section 2.12 – each sample was analysed in three independent repetitions! Please clarify.
Author response: Thank you for comprehensive review of our MS. Our manuscript was corrected by MDPI Language Editing Service. We also corrected section 2.12, and we supplemented section 2.2 with the explanation that we used 3 bags of mushrooms substrate per every treatment (kind of salt x concentration).
Abstract: Overall it is difficult to grasp when in the production stage of the mushroom that the supplementation with Mg occurred
Author response: The substrate was supplemented with Mg before inoculation. We assumed that Mg was absorbed by mycelium during its growth in the substrate (approx. 3 weeks) and partially transported to fruiting bodies during its growth in subsequent flushes. In commercial production, the oyster mushroom fruiting bodies are harvested in clusters, when they are of the best market quality (appropriate weight and shape), despite of the flush. That’s why we aimed to investigate the Mg content in fruiting bodies from whole harvest period. Anyway, we thank you for this comment, we consider the investigation of fruit bodies chemical composition in following flushes. We added the detailed information on the experiment layout and harvesting period.
Line 24: Please specify % decrease in productivity
Author response: We supplemented abstract with a sentence: “Substrate supplementation with 4200 mg of Mg caused the most significant decrease in mushroom productivity, of about 28% for both Mg salts.”
Line 25: Please specify the lowest concentration of MgCl2 causing a reduction in productivity
We added the abovementioned sentence and we deleted the information on the lowest concentration of MgCl2.
Lines 25-26: This sentence is not clear “but the effect on dry matter content in fruiting bodies was slight but significant”
Author response: We supplemented abstract with a sentence: “The dry matter content in fruiting bodies was significantly lower in treatment with 210 mg of Mg was applied as MgSO4 in comparison to the control.”
Line 26: Please specify % increase in Mg content of the mushroom
Author response: We supplemented abstract with a sentence: “Supplementation effectively increased Mg content in fruiting bodies of P. djamor of 19-85% depending on a treatment...”
Lines 26-27: This sentence is not clear “modified in a differentiated way”
Author response: We replaced “modified in a differentiated way” with “affected significantly”
What does it mean by “pro health effects”
Author response: We deleted “pro health effects”
Lines 56 and 58: Remove the dash/hyphen for “mycelium-had” and “mice – this”
Author response: Thank you for pointing of this error, we corrected it.
Lines 65-66: This sentence is not clear “Although this species is less important than grey oyster mushroom (P. ostreatus) considering the economical approach”
Author response: Thank you for this suggestion. We corrected this sentence in MS to be more clear.
Lines 72-75 is too long and difficult to understand
Author response: Thank you for this suggestion. We divided and supplemented this sentence to be more precise.
Please clarify the term “slat-dependent effect”
Author response: We are sorry for this typing error, the proper phrase is “salt-dependent”. We corrected this error.
Line 109: This sentence is not clear “Mentioned magnesium salts we applied”
Author response: We corrected this sentence to be more clear.
Line 119: Please specify the model of equipment used to perform the sterilization
Author response: We added steam sterilizer model and producer.
Line 127: “1.5 h” repeated twice in the same sentence
Author response; Thank you for an indication of this error, we have corrected it.
Line 130: Please specify the model of the cultivation chamber
Author response: The mushrooms cultivation chamber was constructed with commonly available components and equipped in elements necessary to provide the precise conditions for mushroom cultivation (temperature, humidity, air exchange) and the system of controlling of these parameters. It was dedicated to the experimental approach and self-constructed, because there was not available on the market growing chambers dedicated to small scale experiments with mushroom production.
Line 154: Please clarify “reference solution”
Author response: Addressed.
Section 2.6: Since the sample is pink oyster mushroom, was the extract pink in colour – which could have interfere with the vitamin C assay?
Author response: Extracts of Pleurotus djamor had different tint of pink color. We didn’t have problem with precise titration.
Line 225: Please clarify the term “ samples from intestinal juice”
Author response: We have corrected this error and delete mentioned sentence. In this MS we didn’t have to clear samples through membranes, because samples were mineralized.
Section 2.10.1: Please clarify the DAD detection of phenylalanine and phenolics acids were done at which wavelength. Also, please specify the purify of the standards used for quantification.
Author response: DAD detection of phenylalanine and phenolics acids were done at λ=254 nm. Purify of the standards used for quantification was ≥ 99.0%, from Sigma-Aldrich. Both information were added to MS.
Section 2.10.2: Please specify the purity of the standards used for quantification
Author response: Purity of the standards used for quantification was ≥ 99.0%, from Sigma-Aldrich. Additional information was added to MS
Section 2.10.3: The compound detection and quantification is not clearly described here
Author response: We have added more description to the methodology in MS, as above: “Lovastatin was quantitatively analyzed with the help of a calibration curve and with the assumption of linearity of the size of the area tested under the peak relative to the concentration of the this compound standard (purity ≥ 99.0%) used.”
Section 2.11: The purity of thiamine standard needs to be specified
Author response: We have added information to the methodology about purity of standards of thiamine and riboflavin. Thiamine hydrochloride ≥ 99% (HPLC), ≥ 99%, Sigma-Aldrich, (-)-Riboflavin from Eremothecium ashbyii ≥ 98%, Sigma-Aldrich
Section 2.12: The authors present correlation result in the manuscript – please clarify which correlation test was performed here.
Author response: The chapter ‘2.12 Data analysis’ was supplemented with the sentence: The results were also examined for Pearson's correlation coefficient (r) between analyzed parameters.
Table 1 – What is the “detection level” for sulphate?
Author response: The detection level for sulfate was 0,5 mg/L. We have added this information to the MS.
Table 1 – It is not clear why samples supplemented with MgSO2 not analysed for chloride
Author response: The research on determination of chloride and sulfate was carried out for the control sample and it’s results are presented in this paper in table 1. The earlier pilot studies proved that the addition of a given salt does not significantly change the content of other salt (maximum at ± 5%). Therefore, in the samples containing given salt (SO42-), the content of another salt (Cl-) was not tested. In addition, the subject of this study was to evaluate the effect of the addition of a given salt to increase the concentration of a given anion in the fruiting body from the salt used.
It is not clearly described in the manuscript what is the “standard growing condition”
Author response: We described in 2.2 chapter the constant growing conditions (90 ± 3% humidity, 18 ± 2 °C, and photoperiod 12 h of light intensity 11 mmol s-1 m-2) which are recommended for P. djamor. We added also a phrase ‘Standard growing medium’ to 2.2. chapter to clarify substrate composition.
Round 2
Reviewer 2 Report
The authors have addressed all the comments. Therfore, I suggest the acceptance of the manuscript in its present form.
Reviewer 3 Report
Revision was satisfactory and may proceed to publication. Congratulations for the good work :)